# View of Saudi Arabia Strategy for Water Resources Management at Bishah, Aseer Southern Region Water Assessment

Hesham K. Fazel [1], Sayeda M. Abdo [2,*], Atiah Althaqafi [1,3], Saad H. Eldosari [3], Bao-Ku Zhu [4] and Hosam M. Safaa [5,6]

[1] Department of Business Administration, College of Business, University of Bishah, Bishah 67714, Saudi Arabia; hfazel@ub.edu.sa (H.K.F.); aalthaqafi@mewa.gov.sa (A.A.)

[2] Water Pollution Research Department, Environmental Research Division, National Research Center, Dokki, Giza 12622, Egypt

[3] Ministry of Environment, Water and Agriculture, Bishah 67714, Saudi Arabia; sahaljiriah.c@nwc.com.sa

[4] Department of Polymer Science and Engineering, Zhejiang University, Hangzhou 310058, China; zhubk@zju.edu.cn

[5] Department of Biology, College of Science, University of Bishah, Bishah 67714, Saudi Arabia; hosam.safaa@agr.cu.edu.eg

[6] Department of Animal Production, Faculty of Agriculture, Cairo University, Giza 12613, Egypt

* Correspondence: sayedamohammed2015@gmail.com

**Abstract:** Water quality management is critical for the preservation of freshwater resources in semi-arid and arid areas, which are necessary for long-term development. Local authorities and water resource managers can allocate resources for potable or agricultural needs based on the quality of water in various places. A total of 14 water samples were collected and examined in this study. Microbiological, chemical and physical analyses were considered as important indicators for assessing water quality. Physical, chemical, and microbiological data were measured and evaluated as essential markers for determining water quality. A comparison was made between these characteristics and the King Saudi Water Standard (GSO149/2014). According to the findings, results of infiltration pond and Tabla Dam manifest the anthropogenic activities and natural influences of the greatest impact on water quality. Therefore, a reliable assessment approach for assessing water quality is very important for decision makers and for constructing sustainable development plans.

**Keywords:** development plans; water polices; drinking water; water wells; water sanitation

## 1. Introduction

By 2025, approximately 1.8 billion people living in regions or countries will face water shortages, as estimated by United Nations. In addition, two-thirds of the world's residents may face water stress [1,2]. Despite 99.84% of Saudis having access to drinking water [3], Saudi Arabia is classified as one of the most water-scarce countries in the world. The absolute water scarcity is 500 m$^3$/per person, while the share per individual in Saudi Arabia is 89 m$^3$/per capita. In addition, the Kingdom faces different water challenges at planning, operational and management levels; Saudi Arabia has a third of the highest consumption of individual freshwater in the world. Daily consumption of water per capita increased from 227 L in 2009 to 278 L in 2018.

The Kingdom of Saudi Arabia has relied on desalinated water since the 1950s and became the leading producer of desalinated water in the world. About 7.6 million cubic meters is produced daily, accounting for 22% of global production in 2020 [4]. However, this bears enormous pressure on the environment and energy security.

Groundwater is globally considered as an essential resource of the drinking water supply. In both the developed and developing countries, there has been a manifold increase

in the use of groundwater among the populations in rural areas, as well as in the rapidly increase urban areas [5–8]. Due to limited availability and deteriorated quality of surface water in the latter half of the twentieth century, groundwater usage grew rapidly in most countries of the world. Wherever groundwater is devoid of viruses and dangerous bacteria, groundwater quality has significant advantages over surface water. Furthermore, there is a lower organic matter content in ground water [9].

Several anthropogenic fluxes, such as industrial waste discharge, pesticides, excessive use of fertilizers, gas and oil spillage, landfills and mining waste have a substantial impact on groundwater quality. Clean-up is difficult and expensive once groundwater has been contaminated [10]. Groundwater quality status and related health concerns, as well as a study of the factors impacting groundwater quality, are essential for defining policies to protect and manage groundwater quality [11]. Groundwater improvement is considered a key option for solving gaps in action plans and providing flexibility to meet the consequences of climate change in both developed and developing nations [12–14]. Groundwater is already the preferred source of drinking water globally [15]. Ground water is typically regarded more reliable and accessible than surface water since it can be used directly by consumers [16].

Water management rules are critical for promoting the security of water sources, and establishing these necessitates for the collaboration of professionals, government officials, and the general public [17–19]. Furthermore, considering the effects on human health is the first step in ensuring long-term water quality management. In a loess location of northwest China, a study assessed health and groundwater quality concerns from Cr6+ and NO-3 [20,21].

The Kingdom of Saudi Arabia (KSA)water plan vision 2030, as well as the Bishah governorate (one of the southern governorates—Aseer district—KSA) as a model for developing the water system to combat population expansion. The plan's ultimate purpose is to close any gaps in water production, strategic storage, and treatment capacity by calculating the rate of population growth and water needs for the following nine years, then building projects appropriately.

## 2. Materials and Methods

### 2.1. Sampling Sites

Bishah is a town in the southwestern Saudi Arabian province. Bishah was its own province before merging with its neighboring province, Aseer. Bishah has a population of 205,346 according to the 2028 Census [22], with nearly 240 villages and 58 larger settlements that are spread out on both sides of the Bishah Valley (the longest valley in the Arabian Peninsula). The city is located to the south of the Arabian Peninsula and located at 20°0′0″ N 42°36′0″ E. It stands at an altitude of approximately 610 m (2000 ft.) above sea level. The government of Saudi Arabia's strategic plan involves using a newly constructed dam to supply water to population centers.

King Fahad's dam is located on Wadi Bishah at a distance of 40Km south of Bishah governorate. Wadi Bishah is considered as one of the biggest valleys in the Arabian Peninsula where its length exceeds 250 km up to the dam site. It runs from Aseer heights through Aseer hill. Wadi Bishah collects its flow from more than 100 tributaries; among those are Wadi Khamis Mushait, Wadi Abhaand others. Wadi Bishah continues after the dam towards the north along 200 km where a number of tributaries flow into it, such as Wadi Hurgab, Turuj and Tabalah, till its meet with Wadi Tathleeth forming one stream known as Wadi Al-Dawaser which extends as far as 200 Km and ends in Rumeila in Rub' Al Khali (Figure 1).

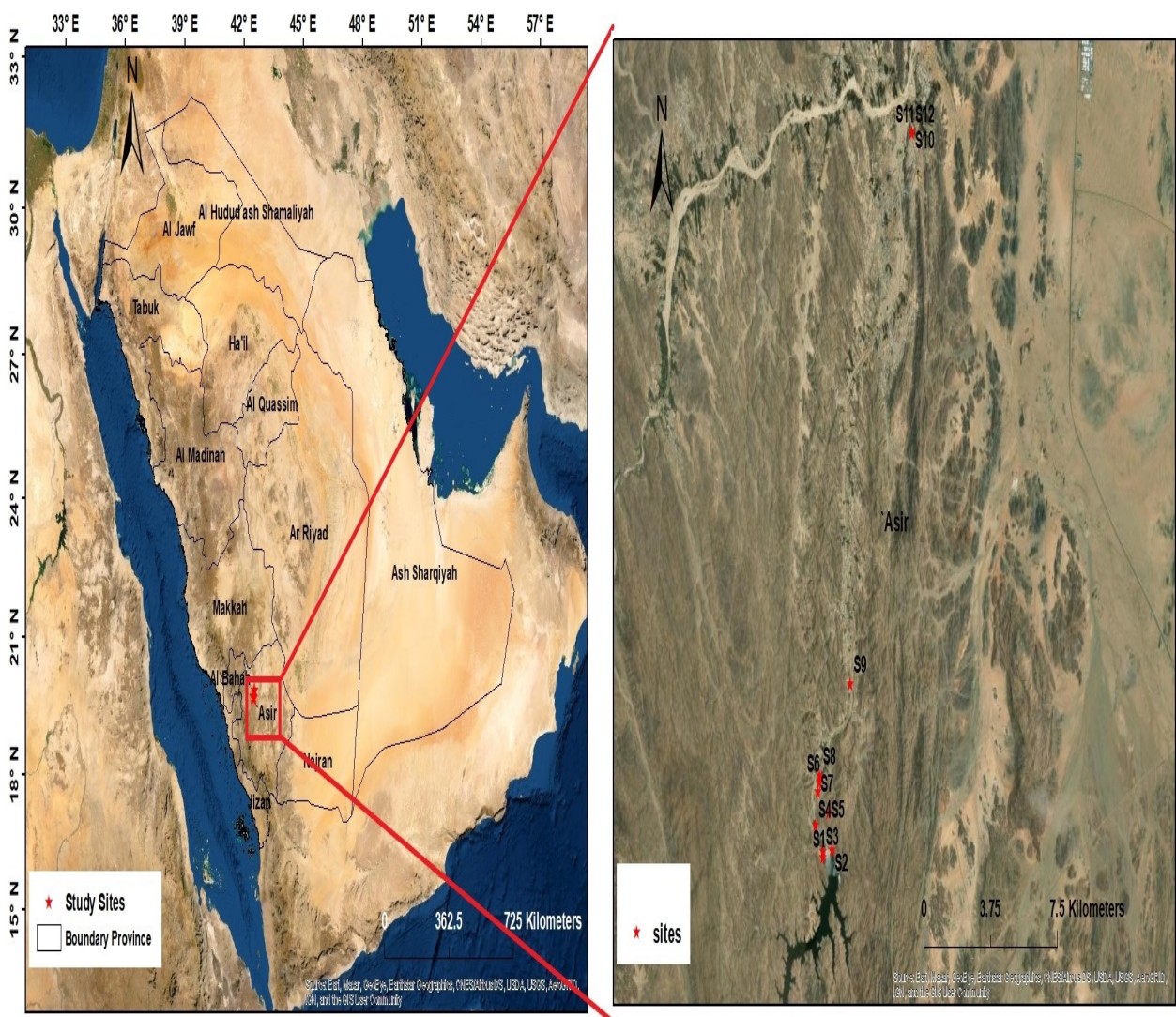

**Figure 1.** Bishah town map and sampling sites (locations).

Monthly water samples were collected from different sites which includes the network of the Bishah town and the extension of the surrounding sources and networks such as: Wadi Bishah (1), King Fahd Lake Dam, Bishah Dam, Bishah Wells (1–5), Well Lake (1), Well Lake (2), Infiltration Pond, King Fahd Station Bishah, Bishah Tank, Wadi Bishah (2), Ashyab King Fahd Dam, Wadi Hergab and Tabala Dam (Pure Water).

*2.2. Water Analysis*

Physical water quality parameters such as temperature, pH, turbidity, and electric conductivity (EC) were measured at time of samples collection. Physical water quality parameters such as temperature (measured on the site using mercury thermometer), pH (measured using digital pH meter-Model Metrohm, pH Lab 827), turbidity (measured by Nephelometer using NTU standards), electric conductivity (EC) (measured using Salinometer (Thermo Electron Corporation, model: Orion 150A+, (United States of America (USA)) were measured at time of samples collection.

The water samples were stored in an ice box and then transported to the lab. Simultaneously, water samples were collected for microbiological analysis also, stored in icebox and the samples fixed by the addition of sodium thiosulphate crystals. Total Coliforms and *E. coli* were measured by Most Probable Number (MPN) method [23].

The standard analytical procedures described in APHA [23] were followed for chemical water sample analysis. Total dissolved solids (TDS), total alkalinity, total hardness (TH),

calcium hardness (Ca H) and magnesium hardness (Mg H), chloride, sulphate, ammonia, nitrite, nitrate, iron, manganese, fluoride were measured.

The digested water samples were then analyzed for Lead (Pb), Cadmium (Cd), Chromium (Cr), Nickel (Ni), Copper (Cu), Cobalt (Co), Aluminum (Al), Arsenic (As), Selenium (Se), Barium (Ba), Beryllium (Be), Boron (B), Molybdenum (Mo), Vanadium (V), Lithium (Li), and Zinc (Zn) using a Flame Atomic Absorption Spectrophotometer AAS model (Model: AA-6800 Shimadzu, Japan). The metal concentrations in each digested sample were determined by comparing their absorbance with aqueous calibration standard prepared from the stock standard solutions of the respective elements [23].

To prevent contamination, all materials associated with trace metal sampling and analyses were thoroughly acid cleaned before use. Glassware and Teflon vessels were treated in a solution 10% $v/v$ nitric acid for 24 h and then washed with distilled and deionized water.

### 2.3. Statistical Data Analysis

The data were presented in tables as arithmetic means, and the standard deviation (SD) was computed [24]. All statistical visualizations were performed by GraphPad Prism 8.3.0 (GraphPad, San Diego, CA, USA).

### 3. Results and Discussion

The Water Strategy 2030, which is in line with the new vision or strategic direction of the water sector, is responsible for the development of water policies in the Kingdom of Saudi Arabia. The plan's ultimate goal aims to fill any gaps in water production, storage strategy, and treatment capabilities by analyzing supply and demand over the next seven years and planning future projects. As a result, the amount of future water needed to supply municipal water in the Aseer region (Figure 2) is estimated to be around 449,000 m$^3$/day (2030).

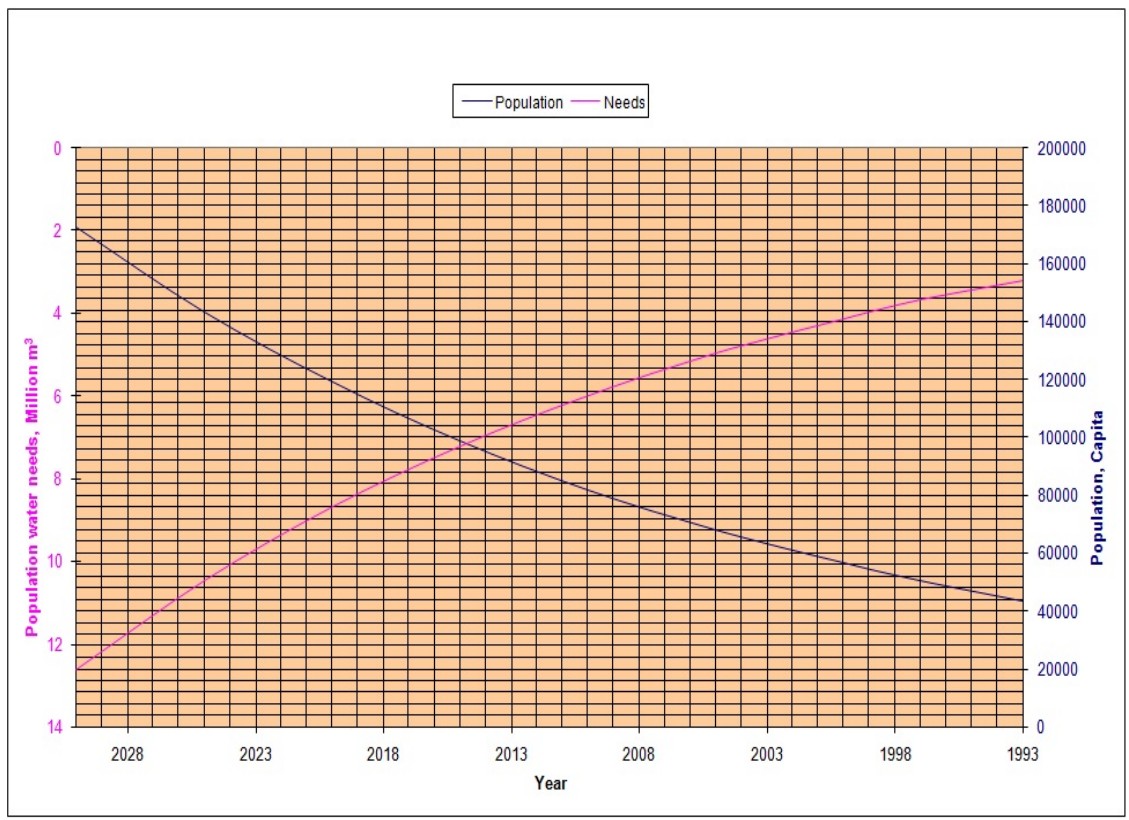

**Figure 2.** Bishah population and water requirements till 2028.

The renewable water resources, on the other hand, in the Aseer region that can be exploited are assumed to be 109,000 m$^3$/day (2015–2019). Therefore, a shortage of municipal water demand will arise in the planned target year only through the development of renewable water resources [25,26]. The Ministry of Environment, Water and Agriculture [27] constructed a dam (King Fahad Dam) in the Bishah Valley to support agricultural activities in the region. The dam flow has also been redirected for drinking purposes due to the paucity of pure drinking water.

Furthermore, efforts to deliver water must be accompanied by ensuring its quality according to Standard levels of water quality traits according to WHO and GSO 149 (Table 1). Thus, the mean results of the in situ tests and laboratory analysis of water samples from 14 sampling sites in Bishah town along the period of 2006–2016 were shown in Tables 2 and 3. Moreover, Figures 3 and 4 revealed minimum, maximum and mean readings of tested samples.

**Table 1.** Standard levels of water quality traits according to WHO (2011a) and GSO 149 (2014).

| Parameters | Unit | WHO [28] | GSO [29] |
|---|---|---|---|
| pH | | 6.5–8.5 | 6.5–8.5 |
| Color | | 15 | 15 |
| Turbidity | NTU | 5 | 5 |
| Total Dissolved Solids (TDS) | mg/L | 1000 | 1000 |
| Total alkalinity (CaCO$_3$) | mg/L | | |
| Total hardness (CaCO$_3$) | mg/L | 500 | 500 |
| Chloride | mg/L | 250 | 250 |
| Sulphate | mg/L | 250 | 250 |
| Ammonia | mg/L | 1.5 | 1.5 |
| Nitrite | mg/L | 3 | 3 |
| Nitrate | mg/L | 50 | 50 |
| Fluoride | mg/L | 1.5 | 1.5 |
| Iron | mg/L | 0.3 | 0.3 |
| Total coliform | MPN/100 mL | -ve | -ve |
| *E. coli* | MPN/100 mL | -ve | -ve |

**Table 2.** Water quality parameters of Bishah water resource locations.

| Parameter, SD | Unit | Wadi Bishah (1) | King Fahd Lake Dam | Bishah Dam | Bishah Wells (1–5) | Well Lake (1) | Well Lake (2) | Infiltration Pond | King Fahd Station, Bisha | Bishah Tank | SD |
|---|---|---|---|---|---|---|---|---|---|---|---|
| pH | | 7.20 | 7.98 | 8.47 | 8.10 | 8.33 | 9.70 | 8.40 | 7.00 | 7.62 | 0.721 |
| Color | Co/Pt unit | 5 | 4 | 5 | 4 | 4 | 5 | 6 | 4 | 4 | 2.42 |
| Turbidity | NTU | 5.70 | 3.86 | 5.51 | 0.10 | 5.32 | 2.80 | 5.00 | 0.60 | 0.45 | 1.275 |
| TDS | mg/L | 716 | 391 | 334 | 495 | 386 | 202 | 450 | 535 | 367 | 225.1 |
| Total alkalinity (CaCO$_3$) | mg/L | 141 | 80 | 111 | 165 | 110 | 81 | 119 | 124 | 138 | 44.3 |
| Total hardness (CaCO$_3$) | mg/L | 351 | 207 | 160 | 221 | 155 | 131 | 157 | 218 | 224 | 83.3 |
| Calcium hardness (CaCO$_3$) | mg/L | 275 | 132 | 112 | 168 | 124 | 63 | 109 | 164 | 156 | 60.7 |
| Magnesium hardness (CaCO$_3$) | mg/L | 76 | 75 | 48 | 53 | 35 | 35 | 48 | 54 | 61 | 22.7 |
| Calcium | mg/L | 110 | 53 | 45 | 67 | 50 | 25 | 44 | 66 | 62 | 21.67 |
| Magnesium | mg/L | 18.3 | 17.9 | 11.5 | 12.7 | 8.4 | 8.4 | 11.5 | 13 | 15 | 5.43 |
| Chloride | mg/L | 192 | 104 | 63 | 84 | 114 | 33 | 84 | 102 | 77 | 47.2 |
| Sulphate | mg/L | 197 | 107 | 87 | 87 | 97 | 63 | 64 | 90 | 67 | 43.7 |
| Ammonia | mg/L | 0.01 | 0.01 | 0.01 | 0.34 | 0.20 | 0.02 | 0.17 | 0.01 | 0.01 | 0.082 |

**Table 2.** *Cont.*

| Parameter, SD | Unit | Wadi Bishah (1) | King Fahd Lake Dam | Bishah Dam | Bishah Wells (1–5) | Well Lake (1) | Well Lake (2) | Infiltration Pond | King Fahd Station, Bisha | Bishah Tank | SD |
|---|---|---|---|---|---|---|---|---|---|---|---|
| Nitrite | mg/L | 0.01 | 0.01 | 0.01 | 0.07 | 0.01 | 0.01 | 0.04 | 0.02 | 0.01 | 0.047 |
| Nitrate | mg/L | 0.44 | 0.24 | 0.30 | 0.18 | 0.25 | 0.87 | 0.43 | 0.33 | 0.31 | 2.65 |
| Fluoride | mg/L | 0.73 | 1.10 | 1.97 | 3.00 | 1.40 | 0.73 | 9.00 | 3.00 | 0.98 | 0.105 |
| Iron | mg/L | 0.08 | 0.04 | 0.09 | 0.04 | 0.04 | 0.10 | 0.12 | 0.07 | 0.04 | 0.044 |
| Total coliform | MPN/100 mL | -ve | -ve | -ve | -ve | -ve | -ve | +ve | -ve | -ve | |
| *E. coli* | MPN/100 mL | -ve | -ve | -ve | -ve | -ve | -ve | +ve | -ve | -ve | |

Means are for samples collected during the period of the study (2006–2015), 30 samples per parameter/location/year. Since 2016, Al Wajid drinking water system was applied and these resources were used as alternative ones. SD = Standard deviation; TDS = Total Dissolved Solids. -ve = negative; wells (1–5) collection of 5 wells.

**Table 3.** Water quality parameters of Bishah water resources' locations feed other water supply lines.

| Parameter | Unit | Wadi Bishah (2) | Bishah Tank | Ashyab King Fahd Dam | Wadi Hergab | Tabala Dam (Pure Water) | SD |
|---|---|---|---|---|---|---|---|
| pH | | 8.23 | 7.62 | 8.11 | 7.90 | 8.20 | 0.656 |
| Color | Co/Pt unit | 4 | 4 | 7 | 7 | 4 | 2.42 |
| Turbidity | NTU | 4.54 | 0.45 | 0.84 | 5.30 | 7.82 | 1.275 |
| TDS | mg/L | 564 | 367 | 469 | 140 | 129 | 125.1 |
| Total alkalinity (CaCO$_3$) | mg/L | 112 | 138 | 155 | 64 | 88 | 44.3 |
| Total hardness (CaCO$_3$) | mg/L | 316 | 224 | 272 | 102 | 96 | 76.3 |
| Calcium hardness (CaCO$_3$) | mg/L | 254 | 156 | 228 | 88 | 85 | 59.7 |
| Magnesium hardness (CaCO$_3$) | mg/L | 63 | 61 | 44 | 14 | 11 | 22.7 |
| Calcium | mg/L | 102 | 62 | 91 | 35 | 34 | 23.67 |
| Magnesium | mg/L | 15.1 | 14.7 | 10.6 | 3.4 | 2.6 | 5.33 |
| Chloride | mg/L | 148 | 77 | 124 | 15 | 8 | 47.2 |
| Sulphate | mg/L | 180 | 67 | 132 | 22 | 12 | 47.7 |
| Ammonia | mg/L | 0.03 | 0.01 | 0.01 | 0.19 | 0.19 | 0.091 |
| Nitrite | mg/L | 0.01 | 0.01 | 0.01 | 0.01 | 0.01 | 0.052 |
| Nitrate | mg/L | 0.36 | 0.31 | 0.26 | 0.13 | 0.21 | 2.73 |
| Fluoride | mg/L | 0.43 | 0.98 | 0.80 | 0.80 | 0.80 | 0.108 |
| Iron | mg/L | 0.09 | 0.04 | 0.02 | 0.17 | 0.26 | 0.053 |
| Total coliform | MPN/100 mL | -ve | -ve | -ve | -ve | +ve | |
| *E. coli* | MPN/100 mL | -ve | -ve | -ve | -ve | +ve | |

Values are means of 30 collected samples each per parameter/location. SD = Standard deviation; TDS = Total Dissolved Solids.

Tables 2 and 3, Figures 1 and 2 show the findings of in situ tests and lab analysis of water samples collected from 14 sampling locations in Bishah town from 2006 to 2016 (Figures 3 and 4).The pH levels in drinking water are in the range of 7–9.7 (mean = 8.1), according to the specifications of drinking water quality requirements.

Figures 3 and 4 show the minimum, lowest, mean value, and standard division for each parameter and a pH level of 9.7 was observed in samples obtained from the Well lake (2) sampling site, measuring above the WHO [28] and GSO149 [29], pH requirements (6.5–8.5).

The pH level is significant since it influences a number of solubility and geochemical reactions in groundwater. Furthermore, in a treatment plant, pH is an important operational parameter. For chemical treatments including coagulation, disinfection, softening, and corrosion control, the pH must be kept within a safe range. Failure to keep corrosion to a minimum can lead to contamination of drinking water and aesthetics [30].

In addition to turbidity and odor, which are routinely examined, the color of potable water is one of the aesthetic quality characteristics [31].

Drinking water should, in theory, be colorless. The presence of colored organic matter (mainly fulvicandhumic acids), iron and other metals, in addition to contamination of the water supply with industrial effluents can all cause drinking water coloration.

Accordingly, all sites under investigation meet the standards of the WHO (ref. [28] with all readings under the permissible level and maximum, minimum and mean values reaching 3.6, 7 and 4.8 (SD $\pm$ 1.24) Co/Pt unite (Figures 3 and 4a).

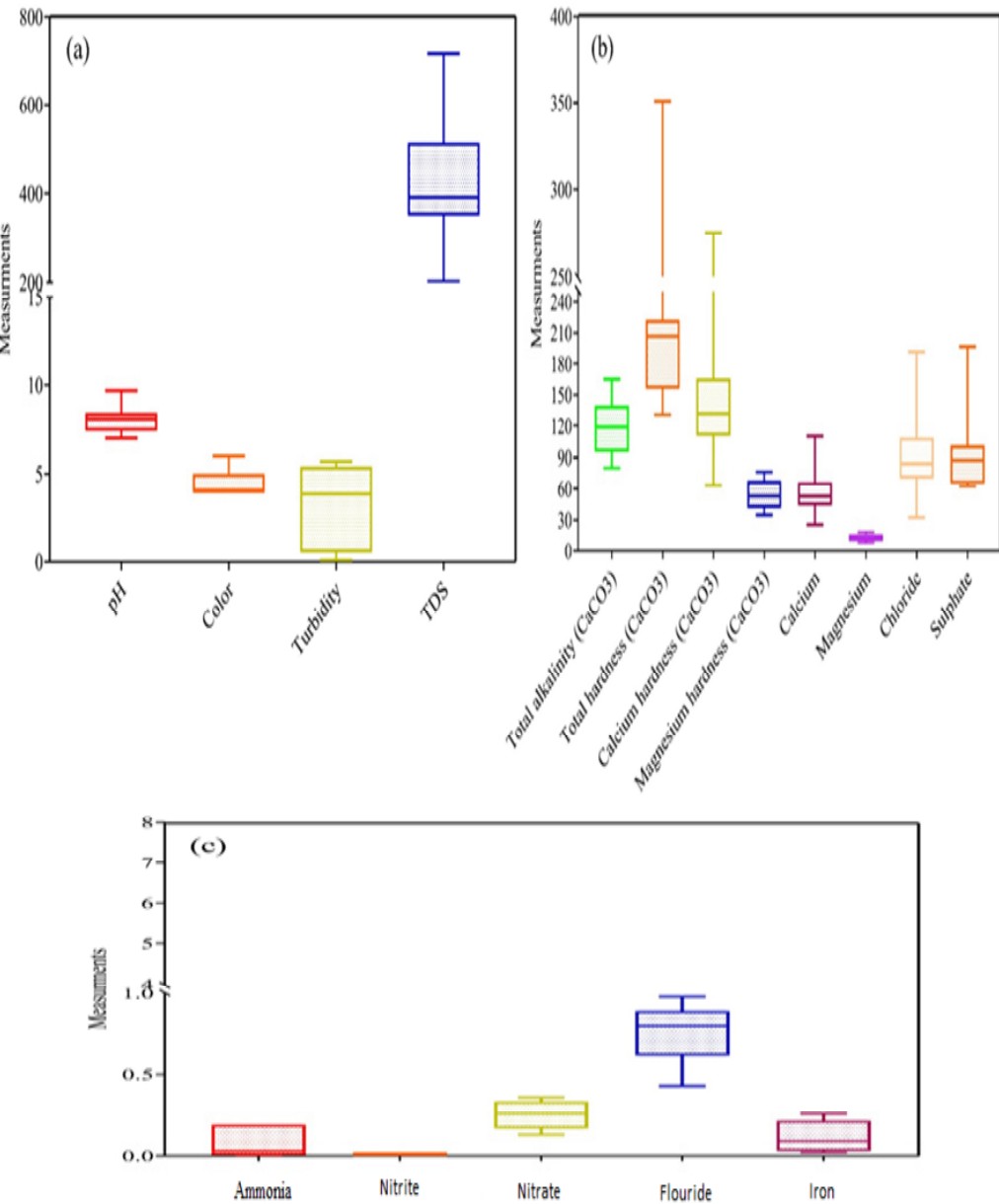

**Figure 3.** (**a–c**) Water quality parameters (maximum, minimum and mean values) of Bishah water resources.

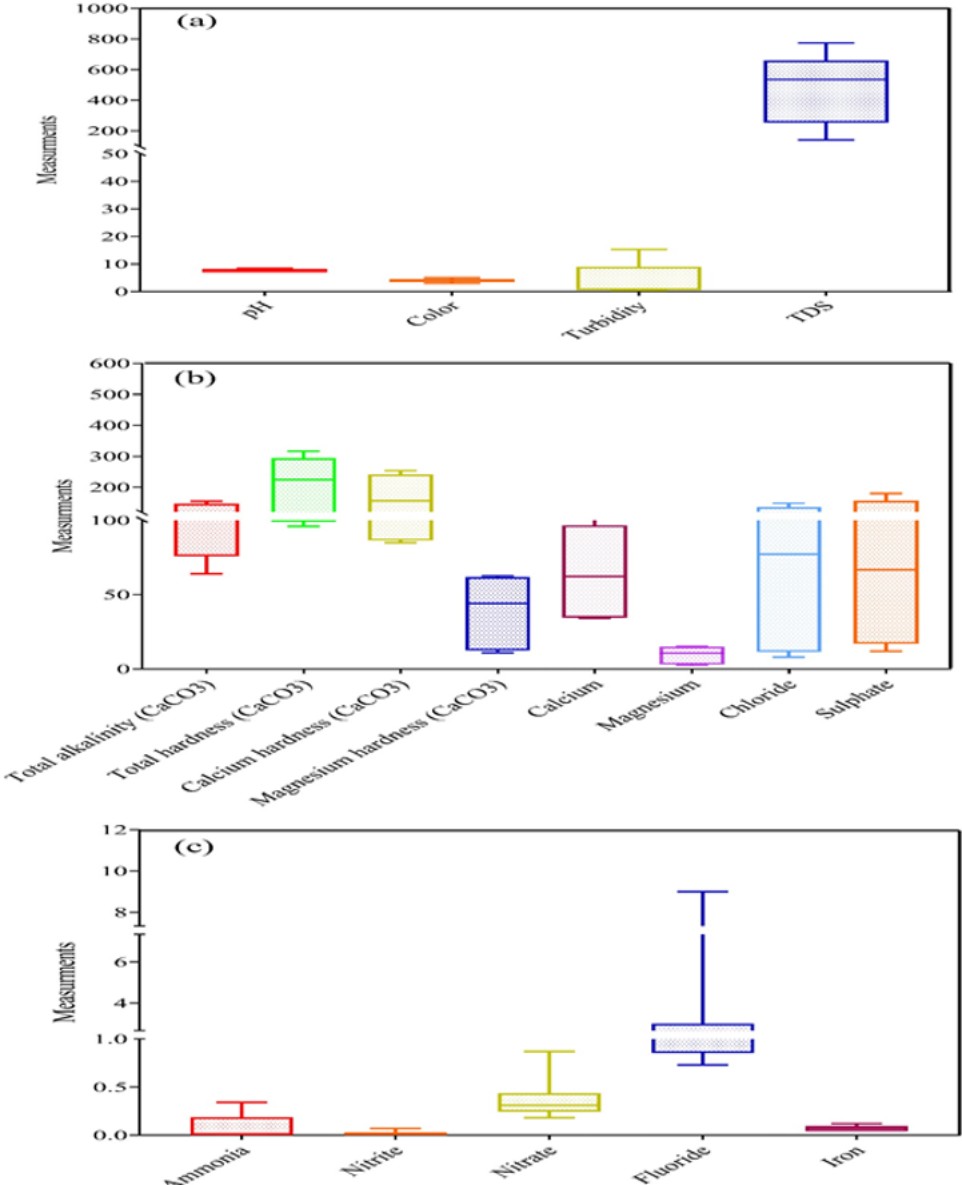

**Figure 4.** (**a**–**c**) Water quality parameters (maximum, minimum and mean value) of Bishah water resources feed other water supply lines.

The turbidity of high-quality drinking water should be minimal. GSO149 [29] recommends a maximum allowed turbidity of less than 5 NTU for drinking water. A variety of samples collected from various sampling locations exhibited turbidity values ranging from 0.1 to 7.82 NTU. The turbidity value in five areas investigated (Tables 2 and 3) exceeded the allowed limits (5 out of 14). For aesthetic reasons, turbidity levels in drinking water are significant as well as treatment plant performance, as extreme turbidity can shield harmful germs from disinfectant actions, making water filtration more difficult and expensive [30,32].

There are no actual health requirements for main ions or TDS in drinking water; however, high amounts of calcium and magnesium can cause scaling, and excessive salinity might affect odor and taste [33,34]. The TDS levels ranged from 129 to 716 mg/L, with a mean value of 98 mg/L. Collected samples were acceptable for human health, according to TDS standard limits (1000 mg/L) [35]. Furthermore, drinking water quality can be classified as excellent (300 mg/L), good (300–600 mg/L), fair (600–900 mg/L), poor (900–1200 mg/L), and undesirable (>1200 mg/L) based on TDS [28].

$Ca^{2+}$ deficiency in the human body causes a variety of illnesses, including stroke, osteoporosis, and colorectal cancer. High $Mg^{2+}$ content in drinking water exerts a laxative effect [28,35]. The calcium and magnesium hardness concentrations in this study are within acceptable limits (Tables 2 and 3).Furthermore, bicarbonate concentrations (reported as total alkalinity as $CaCO_3$) ranged from 64 to 165 mg/L. In addition, total hardness concentration as $CaCO_3$ varies from 96–351 mg/L in studied locations, with a mean value of 200.7 mg/L.

All TH, Ca H, and Mg H levels were within the WHO's [36] and GSO149 [29] recommended tolerable limits of 500, 350, and 150 mg/L, respectively. The water quality of all samples was classified as freshwater (TDS) or very hard water [37,38]. Hard water is also not a health hazard below the permissible level, although it can influence the acceptability of drinking water [28] by causing scale deposition in heated water applications and in water distribution system [36].

The anion chloride (Cl) and sulphate ($SO_4$) are two of the most frequent anions found in water supplies. The concentration of sulphate in the locations examined ranges from 12 to 197 mg/L. These levels were well below maximum permissible limit (250 mg/L, according to WHO [28] and GSO149 [29]. Sulphate is not a health hazard in drinking water at the maximum permitted range [28], although it can have a laxative effect at high levels, leading to intestinal pain and dehydration.

In addition, the concentration of chloride ion (Cl) ranged from 8 to 192 mg/L.

The maximum permitted chloride concentration in drinking water is 250 mg/L.

Chlorine and sulphate levels were all within the acceptable range. Furthermore, the mean value and standard division were, respectively, 88.31 51.99 and 92.65.4. (Figures 3 and 4). Moreover, people are unaffected by chloride; however, at concentrations greater than 250 mg/L, it imparts a salty flavor to water, which many people find undesirable [28].

Excessive concentrations of chloride can also affect metal corrosion in water distribution system pipes, potentially raising metal concentrations in drinking water [39]. Furthermore, an increase of chloride concentration in groundwater is commonly used as a measure of contamination [40].

High levels of ammonia, nitrite, and nitrate are undesirable in drinking water because they provide health risks, particularly to pregnant women and infants. Additionally, groundwater pollution is primarily caused by ammonia and nitrate, such as manure leaching or fertilizer, septic tank leaks, wastewater disposal and so on. The presence of high concentration of ammonia and nitrite is a signal for the existence of other dangerous contaminants such as bacteria or pesticides [41–44]. Ammonia (0.0–0.34), nitrite (0.01–0.07) and nitrate (0.13–0.87) concentrations are in the allowable range of WHO [39] and GSO149 [29] (Tables 2 and 3). The mean values of ammonia, nitrite and nitrate are in the range of 0.018 to 0.33 (Figures 3 and 4) with standard deviation range of ±0.018–0.185.

Fluoride (F−) at low concentrations is necessary for human health, but causes endemic fluorosis (dental and skeletal) at high levels and damage to the soft tissues (kidney, liver, lung, testis, etc.) [45,46]. The studied locations samples revealed fluoride concentration in the range of 0.43 to 9mg/L. Since 4 of 14 sample locations (namely Bishah dam, Bishah wells, infiltration pond and King Fahd station) were out of the standard permissible concentration (1.5mg/L WHO [28] GSO149 [29]). Iron levels were obtained ranging from 0.04–0.26 with a mean value of 0.089 ± 0.066.

Heavy metals in drinking water are a threat to human health considering their strong toxicity even at very low concentrations. Accumulation of few trace elements in the human body may induce cancer [47]. At higher doses, it can cause irreversible brain damage and in extreme cases, death [48].

Adverse effects and toxicity levels depend on heavy metal species and its concentration. The adverse effects include development of autoimmunity, nervous system damage, reduced growth and kidney or liver damage. The mean concentration (mg/L) of the trace elements in descending order was as follows: Be (0.425) >B (0.395) > Al (0.119) > Ba (0.043) > Zn (0.007) >V (0.005) > Mo and Se (0.004) > Cu and Pb (0.003) > Cr and As (0.002) >

Cd (0.00). All trace elements did not exceed the allowable limits of (Table 4) (WHO [28], GSO149 [29]).

**Table 4.** Trace elements composition of water in the study region according to standards.

| Trace Elements | Min. | Max. | Mean | SD | WHO [28] | GSO [29] |
|:---:|:---:|:---:|:---:|:---:|:---:|:---:|
| Al | 0.035 | 0.200 | 0.119 | 0.058 | 0.200 | 0.200 |
| Cr | 0.001 | 0.050 | 0.002 | 0.015 | 0.050 | 0.050 |
| As | 0.001 | 0.010 | 0.002 | 0.003 | 0.010 | 0.010 |
| Se | 0.001 | 0.100 | 0.004 | 0.030 | 0.010 | 0.010 |
| Zn | 0.001 | 3.000 | 0.007 | 0.947 | 3.000 | 3.000 |
| Cd | 0.001 | 0.003 | 0.000 | 0.001 | 0.003 | 0.003 |
| Ba | 0.032 | 0.700 | 0.043 | 0.208 | 1.300 | 0.700 |
| Pb | 0.001 | 0.010 | 0.003 | 0.003 | 0.010 | 0.010 |
| B | 0.286 | 0.500 | 0.395 | 0.069 | 2.400 | 0.500 |
| Mo | 0.002 | 0.070 | 0.004 | 0.021 | 0.070 | 0.070 |
| Cu | 0.001 | 1.000 | 0.003 | 0.315 | 2.000 | 2.000 |
| Ni | 0.009 | 0.167 | 0.033 | 0.048 | 0.070 | 0.070 |

Means are of 30 collected samples each per parameter. Min. = Minimum; Max. = Maximum; SD = Standard deviation.

Escherichia coli (*E. coli*) is a Gram-negative bacteria that resides in the intestine of warm-blooded animals. To verify water quality and ensure there is no biological contamination, *E. coli* is used as an indicator. Detection of *E. coli* in drinking water indicates water has been contaminated with feces containing pathogens. Pathogens can cause many diseases including cholera, typhoid, viral hepatitis A, diarrhea and dysentery [28]. The count of *E. coli* exceeded the maximum allowable limit in 2 locations of 14 studied sites (namely infiltration pond (Table 2) and tabala dam (Table 3)).

The result assessment indicated that regarding the physical–chemical character of water collected from sampling sites under investigation 10 locations were not compliant with WHO 2011 and GSO149/2014.More specifically, one site mismatched due to the pH value (well lake 2), five sites mismatched due to turbidity ((Wadi Bishah (1), Bishah Dam, well lake (1), Wadi Hergab and Tabala Dam) and four sites mismatched due to high fluoride concentrations (Bishah Dam, Bishah wells, infiltration pond and King Fahd Station).Thus, from the microbiological perspective, 2sites (infiltration pond and Tabala Dam) are not compliant with the WHO 2011 and GSO149/2014.

## 4. Conclusions

The groundwater network in Bishah Governorate (One of the southern governorates— King of Saudi Arabia) is influenced by a variety of factors, the most important of which is the quality of the water source. All physical, chemical, and microbiological properties of water samples meet WHO and GSO149 standards. Some samples failed to meet acceptable national and international standards. As a result, the study recommends that:

- groundwater levels be monitored (quantitative and qualitative) on a regular basis
- appropriate pollution control measures be implemented to ensure long-term groundwater quality.
- A need to monitor other pollution indicators, such as organic matter and agricultural wastewater content that leak into groundwater.

**Author Contributions:** H.K.F.: Conceptualization, Funding, writing—original draft. S.M.A.: Methodology, validation, writing—review and editing. A.A.: Conceptualization, methodology, visualization, writing—review and editing. S.H.E.: Software, review and editing. B.-K.Z. Conceptualization, Investigation, methodology, writing review. H.M.S.: data curation, writing. All authors have read and agreed to the published version of the manuscript.

**Funding:** Deputyship for Research and Innovation, Ministry of Education in Saudi Arabia for funding this research work through the project number (UB-06-1442).

**Informed Consent Statement:** Informed consent was obtained from all subjects involved in the study.

**Data Availability Statement:** The datasets used and analyzed during the current study are available from the corresponding author upon reasonable request.

**Acknowledgments:** The authors extend their appreciation to the Deputyship for Research and Innovation, Ministry of Education in Saudi Arabia for funding this research work through the project number (UB-06-1442).

**Conflicts of Interest:** The authors declare no conflict of interest.

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
