# Peer review of "View of Saudi Arabia Strategy for Water Resources Management at Bishah, Aseer Southern Region Water Assessment"

_sustainability, doi:10.3390/su14074198_

Round 1

Reviewer 1 Report

The article is at a low level. It is the usual analytical test of water from various sources. I don't understand this methodology. Did the research concern raw water (well, lake, etc.) or tap water? If they related to raw water, some of the indicators exceeding the norms will be corrected in the water treatment plant. However, if the tests concerned water from the water supply system, some indicators disqualify it. Maybe a lot will be explained if the authors improve the methodology - the last sentence in chapter 2.1 is unfinished.

Detailed comments.

  1. The abstract should end with a short statement - which was a scientific achievement in the research described in this article.
    a. Unfortunately, in this case, we have the obvious " According to the findings, anthropogenic activities and natural influences have the greatest impact on water quality" statement, which could have been written without this research.
    b. postulative sentence "Therefore, reliable assessment approach for assessing water quality is very important for decision makers and for sustainable development plans". It is not known to whom this postulate is addressed.
  2. Fig. 1 - descriptions of vertical axes should probably be the opposite.
  3. The graphs in Figures 3a, 3b, 4a and 4b are unclear. How to understand the logarithmic vertical axis? The scale is logarithmic, but the values are not logarithmic.
  4. Due to the fact that tables 1, 2 and 3 with the results are attached, the graphs in Figs. 3 and 4 are redundant, especially since it is difficult to read anything from them.
  5. Page 4; 22 line from the bottom - it's "standard division" rather it should be "standard deviation".
  6. Page 5; 2nd line from the top - it is "unite", rather it should be "unit".
  7. It is not clear how the research was done. How many samples were repeated from the same sampling sites. If these 14 tests were performed from different locations (the water was different), the calculation of the mean and standard deviation is not correct.
  8. WHAT CONCLUSIONS ARE THE AUTHORS TAKING FROM THIS RESEARCH?

Reviewer 2 Report

Please find the reviewer's comments in the "commets section" of the attached pdf.

It has got potential, but the feeling while reading it was as if it was "written in a hurry": The article needs major improvents regarding the presentation of the results, the figures and the readability of the text.

Round 2

Reviewer 1 Report

  1. What is the reason for variable numbering of tables? Table (A), Table 1, Table 2 and Table 3.
  2. Table (A) - Why are "blank lines" included? Alkalinity, calcium hardness, magnesium hardness, calcium, magnesium etc. are not normalized so they should not appear in the table.
  3. Fig. 4c - the vertical axis is incorrectly scaled. It should have a range of 0-2 or even 0-1. With the current scale it is not possible to read any values from the graph.
  4. Conclusion is a repetition of some sentences. It would be useful to make some important conclusion. The suggestion that water quality and quantity should be monitored is too obvious.

Reviewer 2 Report

Please see the comments in the attached PDF.

Figures and tables must be placed close to the explanatory text. This was improved from the first version. However, the new placement has been done like a "tetrix" and it should be improved.

Figures must be improved: The readability and resolution is very poor.

Round 3

Reviewer 2 Report

Figures 1,2,3 and 4 are very poor. They must be re made in quality format and definition.

I would not recommend the publication of this article until figures have sufficient quality.

Author Response

Dear sir,

             I tried to make the figures clear as possible I can. I have improved figures 3 and 4. I hope the quality is better now.

thanks for your time
